# Uveitis and Other Ocular Complications Following COVID-19 Vaccination

**DOI:** 10.3390/jcm10245960

**Published:** 2021-12-19

**Authors:** Elena Bolletta, Danilo Iannetta, Valentina Mastrofilippo, Luca De Simone, Fabrizio Gozzi, Stefania Croci, Martina Bonacini, Lucia Belloni, Alessandro Zerbini, Chantal Adani, Luigi Fontana, Carlo Salvarani, Luca Cimino

**Affiliations:** 1Ocular Immunology Unit, Azienda USL-IRCCS, 42123 Reggio Emilia, Italy; elena.bolletta@ausl.re.it (E.B.); valentina.mastrofilippo@ausl.re.it (V.M.); luca.desimone@ausl.re.it (L.D.S.); fabrizio.gozzi@ausl.re.it (F.G.); chantal.adani@ausl.re.it (C.A.); 2Ophthalmology Unit, Azienda USL-IRCCS, 42123 Reggio Emilia, Italy; danilo.iannetta@ausl.re.it; 3Clinical Immunology, Allergy and Advanced Biotechnologies Unit, Azienda USL-IRCCS, 42123 Reggio Emilia, Italy; stefania.croci@ausl.re.it (S.C.); martina.bonacini@ausl.re.it (M.B.); lucia.belloni@ausl.re.it (L.B.); alessandro.zerbini@ausl.re.it (A.Z.); 4Ophthalmology Unit, DIMES, Alma Mater Studiorum, University of Bologna, S. Orsola-Malpighi Teaching Hospital, 40138 Bologna, Italy; luifonta@gmail.com; 5Rheumatology Unit, Azienda USL-IRCCS, 42123 Reggio Emilia, Italy; carlo.salvarani@ausl.re.it; 6Department of Surgery, Medicine, Dentistry and Morphological Sciences, with Interest in Transplants, Oncology and Regenerative Medicine, University of Modena and Reggio Emilia, 41124 Modena, Italy

**Keywords:** SARS-CoV-2 disease, COVID-19, vaccine, vaccination, uveitis, vaccine-associated uveitis, ocular complications

## Abstract

Coronavirus disease 2019 (COVID-19) vaccines can cause transient local and systemic post-vaccination reactions. The aim of this study was to report uveitis and other ocular complications following COVID-19 vaccination. The study included 42 eyes of 34 patients (20 females, 14 males), with a mean age of 49.8 years (range 18–83 years). The cases reported were three herpetic keratitis, two anterior scleritis, five anterior uveitis (AU), three toxoplasma retinochoroiditis, two Vogt-Koyanagi-Harada (VKH) disease reactivations, two pars planitis, two retinal vasculitis, one bilateral panuveitis in new-onset Behçet’s disease, three multiple evanescent white dot syndromes (MEWDS), one acute macular neuroretinopathy (AMN), five retinal vein occlusions (RVO), one non-arteritic ischemic optic neuropathy (NAION), three activations of quiescent choroidal neovascularization (CNV) secondary to myopia or uveitis, and one central serous chorioretinopathy (CSCR). Mean time between vaccination and ocular complication onset was 9.4 days (range 1–30 days). Twenty-three cases occurred after Pfizer-BioNTech vaccination (BNT162b2 mRNA), 7 after Oxford-AstraZeneca vaccine (ChAdOx1 nCoV-19), 3 after ModernaTX vaccination (mRNA-1273), and 1 after Janssen Johnson & Johnson vaccine (Ad26.COV2). Uveitis and other ocular complications may develop after the administration of COVID-19 vaccine.

## 1. Introduction

Severe acute respiratory syndrome coronavirus 2 (SARS-CoV-2) causes the coronavirus disease 2019 (COVID-19), a multisystemic disorder with medical and socioeconomic consequences that have led to public health crises worldwide. In an effort to alleviate the morbidity and mortality associated with COVID-19 and arrest viral transmission, different types of vaccinations have been developed. Among these vaccines are the inactivated vaccines (PiCoVacc, Sinovac [1]; BBIBP-CorV, Sinopharm [2]), the viral vector vaccines (Ad26.COV2, Janssen Johnson & Johnson [3]; ChAdOx1 nCoV-19/AZD1222, Oxford-AstraZeneca [4]), the messenger ribonucleic acid (mRNA)-based vaccines (BNT162b2, Pfizer-BioNTech [5]; mRNA-1273, ModernaTX [6]), and the protein subunit vaccine (NVX- CoV2373, Novavax [7]).

Common transient local and systemic post-vaccination reactions are pain, redness and/or swelling at the injection site, fatigue, headache, muscle pain, chills, fever, and nausea [4,5]. Although less common, other vaccine-related side effects include cutaneous reactions such as varicella zoster and herpes simplex flares [8].

Different types of ocular complications have also been reported after COVID-19 vaccination, including facial nerve palsy, abducens nerve palsy, new-onset Graves’ disease, episcleritis, anterior scleritis, anterior uveitis (AU), multifocal choroiditis, reactivation of Vogt-Koyanagi-Harada (VKH) disease, multiple evanescent white dot syndrome (MEWDS), acute macular neuroretinopathy (AMN), paracentral acute middle maculopathy (PAAM), thrombosis, and central serous retinopathy (CSR) [9].

The aim of this study was to report uveitis and other cases of ocular complications following COVID-19 vaccination.

## 2. Materials and Methods

This retrospective study included patients with uveitis and other ocular complications following COVID-19 vaccination between January 2021 and October 2021 at the Ocular Immunology Unit, Azienda Unità Sanitaria Locale (AUSL)-IRCCS, Reggio Emilia, Italy.

Data collection consisted of demographic and clinical data. The demographic data included age, sex, general medical and ocular history, and medications. Clinical data included systemic and ocular symptoms post-vaccination, type of vaccine, time interval between vaccination (first and second dose) and symptom onset, laterality of eye disease, ocular findings, treatment, and outcome.

All patients underwent a complete ophthalmic examination with measurement of the best-corrected visual acuity (BCVA), anterior segment slit lamp biomicroscopy, fundus examination, and optical coherence tomography (OCT). Uveitis was graded and classified according to the Standardization of Uveitis Nomenclature (SUN) classification system [10]. In patients with a history of uveitis, the time interval from the last uveitis attack to current uveitis was calculated. In cases of de novo uveitis and no history of uveitis-related systemic disease, laboratory tests were performed at the discretion of the treating ophthalmologist with the aim of excluding other causes of ocular inflammation. These included complete blood count, blood chemistry, erythrocyte sedimentation rate (ESR), C-reactive protein (CRP), Venereal Disease Research Laboratory test (VDRL), Treponema pallidum haemagglutination (TPHA), interferon-gamma release assay (QuantiFERON©-TB Gold test), serum angiotensin-converting enzyme (ACE), serum lysozyme, high-resolution computerized tomography (HRCT) of the chest, and magnetic resonance imaging (MRI) of the brain. In all patients, follow-up was carried out for a minimum of 3 months.

The study and data collection were conducted in agreement with the principles of the Declaration of Helsinki and approved by the local ethics committee (protocol n. 2021/0111389 Comitato Etico Provinciale di Reggio Emilia, Italy). Informed written consent was obtained from all patients.

## 3. Results

### 3.1. Demographics and Clinical Data

The study included 42 eyes of 34 patients (20 females, 14 males), with a mean age of 49.8 years (range 18–83 years). Patients’ demographic data, medical history, type of vaccine, and systemic symptoms post-vaccination are reported in Table 1.

### 3.2. Uveitis and Other Ocular Complications

Mean time between vaccination and ocular complications onset was 9.4 days (median time 7 days, range 1–30 days). Ocular complications reported after the first dose of the vaccine occurred at a mean time of 7.5 days (median time 6.5 days, range 2–30 days), while ocular complications after the second dose of vaccine were reported at a mean time of 10.7 days (median time 7 days, range 1–30 days). Eleven patients had a known history of uveitis and 2 of scleritis, the median time from previous to current attack was 13 months (range 10–108 months). Four patients had a uveitis-related systemic disease: one patient had psoriatic arthritis (PsA), one patient had spondyloarthritis (SpA), and two patients had VKH disease.

The study included three cases of herpetic keratitis, two anterior scleritis, five AU, three toxoplasma retinochoroiditis, two cases of VKH disease reactivation, two pars planitis, two retinal vasculitis, one bilateral panuveitis, three MEWDS, one AMN, five retinal vein occlusion (RVO), one non-arteritic ischemic optic neuropathy (NAION), three activations of quiescent choroidal neovascularization (CNV), and an acute-onset bilateral CSR (Table 2).

Among the three cases of herpetic keratitis, one patient had a previous history of herpetic keratitis and had not been taking systemic antiviral treatment before vaccination. The other two patients that reported herpetic keratitis had a previous history of herpetic keratouveitis and had 1 g of oral valacyclovir once daily for prophylactic therapy during vaccination. The mean time between vaccination and herpetic keratitis onset was 6 days (range 5–7 days).

The five cases of AU included one patient with CMV AU and four with non-granulomatous anterior uveitis (NGAU). Among NGAU were three patients with human leukocyte antigen B27 (HLA-B27) + and three with a previous history of uveitis, two of whom with uveitis-related systemic disease (PsA and SpA) had not been taking systemic therapy at vaccination.

Three patients reported ocular toxoplasmosis, at a mean time of 7.3 days (range 7–8 days) after the vaccination, one patient had an initial episode, and two patients had a recurrence of Toxoplasma retinochoroiditis.

Two patients with bilateral intermediate uveitis had negative laboratory tests and normal chest HRTC and MRI of the brain. Pars planitis was diagnosed in both patients.

One patient presented bilateral panuveitis with retinal vasculitis, papillary oedema, and painful oral ulcers, followed by deep vein thrombosis of a lower extremity. Laboratory workup showed elevated CRP, elevated ESR, and positive pathergy test with a diagnosis of Behçet’s disease (BD).

The female patient who developed AMN 2 days after ChAdOx1 nCoV-19 vaccine was taking combined estrogen–progestin oral contraceptives at the time of vaccination, which were immediately suspended.

Five patients developed RVO: one central retinal vein occlusion (CRVO) and four branch retinal vein occlusion (BRVO). Of these patients, one was affected by diabetes mellitus (DM) and two by systemic arterial hypertension (SAH) (Table 1).

A female patient of 46 years old presented unilateral sectorial papillary oedema, no crowded disc was observed in the other eye. Automatic perimetry demonstrated a diffuse visual field loss. The patient had no clinical signs of giant cell arteritis (jaw claudication, headache, and scalp tenderness), and CRP and ESR were within the normal range. Brain imaging excluded an acute intracranial event, and the patient was diagnosed with NAION. Systemic risk factors associated with NAION such as SAH, DM, hyperlipidemia, and smoking were negative. A hypercoagulable state due to antiphospholipid antibodies was ruled out. Also, anticardiolipin and lupus anticoagulant were negative.

Three cases presented activation of quiescent CNV, secondary to myopia in one patient and secondary to uveitis in the other two. The myopic CNV and one uveitic CNV were never injected, while the other uveitic patient had the reactivation of a lesion treated with intraocular injections of anti-vascular endothelial growth factor (anti-VEGF) drugs 1 year earlier.

### 3.3. Type of Vaccine

Twenty-three cases occurred after Pfizer-BioNTech vaccination (BNT162b2 mRNA), seven after Oxford-AstraZeneca vaccine (ChAdOx1 nCoV-19), three after ModernaTX vaccination (mRNA-1273), and one after Janssen Johnson & Johnson vaccine (Ad26.COV2). Fourteen cases were reported after the first dose of the vaccine and 20 after the second dose (Table 2).

## 4. Discussion

Uveitis and other ocular adverse events have been described following vaccinations for hepatitis B virus (HBV), human papillomavirus (HPV), influenza virus, Bacille-Calmette-Guerin (BCG), varicella virus, measles-mumps-rubella (MMR), yellow fever, hepatitis A virus (HAV), and typhoid [11,12,13,14,15]. Different types of ocular complications have also been reported after COVID-19 vaccination. Pichi et al. reported one patient with episcleritis, two with anterior scleritis, two with AMN, one with PAAM, and one with subretinal fluid soon after receiving an inactivated COVID-19 vaccination (Sinopharm) [16].

A retrospective multicenter study collected 21 cases of unilateral or bilateral AU and 2 cases of MEWDS after the administration of the BNT162b2 mRNA vaccine [17]. Moreover, two case reports described bilateral multifocal choroiditis following COVID-19 vaccination [18,19]. Other cases of facial nerve palsy, abducens nerve palsy, new-onset Graves’ disease, VKH disease reactivation, AMN, PAAM, and thrombosis have been described [9].

This retrospective study reports uveitis and other ocular complications following COVID-19 vaccination. We observed three cases of herpes keratitis reactivation following COVID-19 vaccination.

Vaccines can result in varicella zoster virus (VZV) reactivation, as previously described in patients receiving inactivated vaccines for hepatitis A, influenza, rabies, and Japanese encephalitis. Fernandez-Nieto et al. described 15 cases of herpes simplex/zoster in patients infected with COVID-19 [20]. COVID-19 infection may represent a trigger for herpes reactivation, as recently reported. There are cases reported in the literature characterized by VZV reactivation after vaccination with the mRNA COVID-19 vaccine, including also herpes zoster ophthalmicus (HZO) [21,22,23,24]. It has been postulated that the stimulation of the immune system following vaccination induces a strong T-cell response with increased CD8+ T cell and T helper type 1 CD4+ T cells. Temporarily, VZV-specific CD8+ cells are not capable of controlling VZV after the massive shift of naïve CD8+ cells, which allows VZV to escape from its latent phase. Moreover, another possible explanation focuses on toll-like receptors (TLR) signaling. Abrogations in TLR expression among vaccinated individuals have been linked with marked induction of type I interferon (IFN-I) and potentiation of pro-inflammatory cytokines, which, although they promote T cell immunity and initiate an antibody-secreting memory B cell response, may negatively modulate antigen expression while potentially contributing to VZV reactivation [25].

As reported in the literature, the median time between COVID-19 diagnosis and the development of herpes zoster was 5.5 days [26]. Similarly, the VZV reactivation appeared 5 days after COVID-19 vaccination in a case report [23]. In our three cases, herpes keratitis reactivation occurred after a mean of 6 days (range 5–7 days) after COVID-19 vaccination.

A recent case report of a severe unilateral flare-up of a granulomatous hypertensive uveitis 5 days after the second dose of Moderna vaccine in a patient previously treated for herpes keratouveitis suggests that preventive antiviral treatment should be given in known herpes patients despite quiescent uveitis to avoid potential reactivation [27]. In two of our cases, herpes keratitis reactivation happened in patients with a history of previous herpetic keratouveitis although under systemic antiviral treatment with oral valacyclovir of 1 g once daily.

Episcleritis has been described as ocular manifestations in patients with COVID-19 [28,29,30]. Anterior scleritis has also been reported to manifest after COVID-19 [31]. In addition, scleritis and episcleritis have also been reported in three patients at a mean of 5 days after the first dose of the inactivated COVID-19 vaccine (Sinopharm) [16]. Consistent with the reported literature, our two cases of scleritis were mild and noted at a mean of 5.5 days after vaccination.

In accordance with other studies, we reported AU in patients with or without a history of previous uveitis and/or uveitis-related systemic disease [17,32]. The vaccine-induced increase in IFN-I secretion could potentially drive autoimmune manifestations in patients with a history of autoimmunity or with yet unknown susceptibility to develop one [17,33].

Among our cases, two patients had a recurrence of Toxoplasma retinochoroiditis and one patient an initial episode of Toxoplasma retinochoroiditis at a mean of 7.3 days (range 7–8 days) after the vaccination. The vaccination-induced CD8 T-cell exhaustion may lead to parasite reactivation [34].

Papasavvas I. and Herbort CP. reported a case of VKH disease reactivation 6 weeks after the second dose of the Pfizer anti-SARS-CoV-2 vaccine administration that had been completely under control with a maintenance treatment of infliximab every 10 weeks for 6 years. The patient presented with a severe reactivation of the disease almost as pronounced as during its initial onset [35]. In our study, two patients with VKH disease being treated with mycophenolate mofetil (2 g daily) presented a mild reactivation with choroidal granulomas on indocyanine green angiography (ICGA).

Cases of COVID-19 associated with systemic vasculitis, including retinal vasculitis and papillophlebitis, have been published [36,37]. In our study, two patients presented with bilateral intermediate uveitis and two patients with bilateral retinal vasculitis. These four patients underwent blood tests, chest HRCT, and MRI of the brain, all of which were negative.

A single case of clinical presentation consistent with new-onset BD or a BD-like adverse event following SARS-CoV-2 mRNA-1273 vaccination has been described [38]. Similarly, we reported a case of bilateral panuveitis with new-onset BD.

A multicenter study reported two cases of MEWDS occurring 5 and 30 days after BNT162b2 mRNA vaccination [17]. Our three cases of MEWDS were reported from 4 to 28 days after the first or second dose of BNT162b2 mRNA vaccination.

PAMM and AMN have been reported after H1N1 vaccination. Virgo and Mohamed reported two patients with new paracentral scotoma secondary to AMN and PAMM 16 days after confirmed COVID-19 infections [39]. Furthermore, four case reports described five cases of AMN in young women 2 days after receiving ChAdOx1 nCoV-19 vaccination [40,41,42,43]. Similarly, a case of AMN occurred in a young woman in our study that was taking combined estrogen–progestin oral contraceptives 2 days after ChAdOx1 nCoV-19 vaccination [40,42,43].

Artery or vein retinal occlusion have both been described during or following COVID-19 [44,45], which is thought to induce a systemic inflammatory response, endothelial dysfunction, and a hypercoagulative state, which predisposes patients to systemic thrombus formation [46].

Regarding post-vaccination thrombosis, rare cases of superior ophthalmic vein thrombosis and central retinal vein occlusion have been reported [47,48,49,50]. In this study were five cases of RVO (one CRVO and four BRVO), some of whom were affected by systemic comorbidities including DM or SAH. Moreover, a patient with no ocular or systemic risk factors reported unilateral NAION. So far, four cases of NAION associated with COVID-19 have been described in the literature [51,52,53,54].

Furthermore, three of our patients reported the activation or reactivation of a quiescent CNV secondary to myopia or uveitis

The literature reports a unilateral CSR 3 days after the injection of BNT162b2 mRNA COVID-19 vaccine occurred in a 33-year-old healthy Hispanic male without previous ocular history or pertinent medical history [55]. In our study, a case of acute-onset bilateral CSR in a male patient occurred 13 days after the second dose of BNT162b2 mRNA COVID-19 vaccination.

Most of the patients in our study (58.8%) developed ocular complications after the second dose of the vaccine.

The main limitation of this study was its retrospective design and relatively low number of cases. Previous multiple reports have shown ocular complications following COVID-19 vaccination, although a definitive association can be difficult to demonstrate. However, the close temporal association between vaccination and onset of uveitis or other ocular complications and the similarity to those reported in the literature are quite suggestive.

## 5. Conclusions

COVID-19 vaccination can be followed by herpetic keratitis reactivation in patients with previous herpetic keratitis or kerato-uveitis. The changes in the immune status, including lymphocyte exhaustion, may lead to herpes reactivation [25]. Therefore, prophylactic antiviral therapy with oral valacyclovir, at least for high-risk patients with several previous herpes uveitis episodes, may be considered.

COVID-19 vaccinations can also be followed by anterior scleritis; AU in patients with or without history of previous uveitis, and/or uveitis-related systemic disease; activation of Toxoplasma retinochoroiditis; VKH disease recurrences; pars planitis; retinal vasculitis; panuveitis in new-onset BD, MEWDS, and AMN; as well as RVO (CRVO or BRVO), NAION; activation of quiescent CNV secondary to myopia or uveitis; and CRS.

These complications could be related to the SARS-CoV-2 vaccines’ capacity to induce autoimmune manifestations or thromboembolic events.

Additional epidemiologic and clinical studies and longer follow-up of this cohort are needed to confirm the link between the COVID-19 vaccine and the recurrence or de novo development of uveitis and other ocular complications.

## Figures and Tables

**Table 1 jcm-10-05960-t001:** Patients’ demographics, medical history, type of vaccine, and systemic symptoms post vaccination.

Pt n°	Age	Gender	History of Uveitis	Systemic Disease	Treatment at Vaccination	Time Interval between the Last Uveitis Attack to Current Uveitis (Months)	Type of Vaccine	Systemic Symptoms and Time of Appearance After 1st Dose of Vaccine (Days)	Systemic Symptoms and Time of Appearance After 2nd Dose of Vaccine (Days)
1	83	M	-	-	-		BNT162b2	fever, weakness	-
2	79	M	keratouveitis	-	oral valaciclovir 1g	11	ChAdOx1 nCoV-19	pain at the injection site	pain at the injection site
3	65	F	keratouveitis	-	oral valaciclovir 1g	13	BNT162b2	-	chills, fever, weakness
4	42	F	anterior scleritis	-	-	12	ChAdOx1 nCoV-19	chills, fever	pain at the injection site
5	52	F	anterior scleritis	-	-	13	BNT162b2	weakness, general fatigue	fever, weakness
6	44	M	NGAU (HLA-B27-)	-	-	14	BNT162b2	pain at the injection site	-
7	35	F	NGAU (HLA-B27+)	PsA	-	108	mRNA-1273	fever, weakness	pain at the injection site
8	47	M	NGAU (HLA-B27+)	SpA	-	10	BNT162b2	-	-
9	66	F	-	-	-		ChAdOx1 nCoV-19	pain at the injection site	weakness, general fatigue
10	44	M	-	-	-		BNT162b2	-	pain at the injection site
11	53	M	-	-	-		BNT162b2	pain at the injection site	-
12	58	F	Toxoplasma retinochoroiditis	-	-	82	BNT162b2	pain at the injection site, weakness	-
13	52	F	Toxoplasma retinochoroiditis	-	-	11	Ad26.COV2	fever, chills, weakness	
14	44	F	VKH disease	VKH disease	MMF 2g	22	BNT162b2	pain at the injection site	pain at the injection site
15	58	F	VKH disease	VKH disease	MMF 2g	26	BNT162b2	-	fever, weakness
16	49	F	-	-	-		ChAdOx1 nCoV-19	pain at the injection site	-
17	18	F	-	-	-		BNT162b2	-	fever
18	41	M	-	-	-		mRNA-1273	fever, chills, weakness	pain at the injection site
19	59	F	-	-	-		BNT162b2	-	weakness
20	42	M	-	-	-		BNT162b2	weakness	pain at the injection site
21	53	M	-	-	-		BNT162b2	fever	-
22	18	F	-	-	-		BNT162b2	pain at the injection site	pain at the injection site
23	48	M	-	-	-		BNT162b2	-	-
24	25	F	-	-	estrogen-progestin oral contraceptives		ChAdOx1 nCoV-19	pain at the injection site, fever, muscle pain	
25	39	M	-	-	-		mRNA-1273	fever, chills, muscle pain	pain at the injection site
26	53	F	-	SAH	oral bisoprolol and losartan		ChAdOx1 nCoV-19	pain at the injection site, weakness	-
27	61	F	-	-	-		ChAdOx1 nCoV-19	pain at the injection site	weakness
28	50	M	-	DM	oral metformin		BNT162b2	-	weakness, fever
29	48	M	-	SAH	oral doxazosin		BNT162b2	pain at the injection site	pain at the injection site
30	46	F	-	-	-		BNT162b2	pain at the injection site	-
31	47	F	Toxoplasma retinochoroiditis	-	-	94	BNT162b2	pain at the injection site, chills, fever, weakness	-
32	68	F	Serpiginous Choroiditis	-	-	13	BNT162b2	pain at the injection site, weakness	pain at the injection site
33	66	F	-	-	-		BNT162b2	-	pain at the injection site
34	41	M	-	-	-		BNT162b2	pain at the injection site	weakness, muscle pain

Pt: patient; M: male; F: female; -: none; blank cells: not applicable data; NGAU: non-granulomatous anterior uveitis; HLA-B27: human leukocyte antigen B27; PsA: psoriatic arthritis; SpA: spondyloarthritis; DM: diabetes mellitus; SAH: systemic arterial hypertension; MMF: mycophenolate mofetil.

**Table 2 jcm-10-05960-t002:** Uveitis and other ocular complications post vaccination.

Pt n°	Eye	Ocular Complication	History of Uveitis	Ocular Complication Following 1st or 2nd Dose of Vaccine	Time Interval from Vaccine to Ocular Symptoms Onset (Days)	Ocular Symptoms	BCVA at Presentation (Snellen)	BCVA at Last Follow Up (Snellen)	Treatment Given at Presentation	Outcome
1	LE	herpetic keratitis	-	2nd dose	7	redness, pain, blurred vision	20/40	20/25	acyclovir ophthalmic ointment	complete resolution
2	RE	herpetic keratitis	keratouveitis	1st dose	5	pain, blurred vision	20/40	20/22	oral valaciclovir 1g, dexamethasone eye drops 2 mg/ml	complete resolution
3	LE	herpetic keratitis	keratouveitis	2nd dose	6	redness, blurred vision	20/50	20/20	oral valaciclovir 1g, dexamethasone eye drops 2 mg/ml	complete resolution
4	RE	anterior scleritis	anterior scleritis	2nd dose	6	redness, pain	20/20	20/20	dexamethasone eye drops 2 mg/ml	complete resolution
5	RE	anterior scleritis	anterior scleritis	1st dose	5	redness, pain	20/20	20/20	dexamethasone eye drops 2 mg/ml	complete resolution
6	LE	NGAU (HLA-B27-)	NGAU (HLA-B27-)	1st dose	6	photophobia	20/20	20/20	dexamethasone eye drops 2 mg/ml	complete resolution
7	BE	NGAU (HLA-B27+)	NGAU (HLA-B27+)	2nd dose	1	redness, pain, blurred vision	RE: 20/22 LE: 20/25	RE: 20/20 LE: 20/22	dexamethasone eye drops 2 mg/ml	complete resolution
8	LE	NGAU (HLA-B27+)	NGAU (HLA-B27+)	2nd dose	6	blurred vision	20/25	20/20	dexamethasone eye drops 2 mg/ml	complete resolution
9	LE	NGAU (HLA-B27+)	-	1st dose	30	redness, pain, blurred vision	20/32	20/25	dexamethasone eye drops 2 mg/ml	complete resolution
10	RE	CMV AU	-	2nd dose	8	blurred vision	20/28	20/20	ganciclovir ophthalmic gel 0.15%, dexamethasone eye drops 2 mg/ml	complete resolution
11	LE	Toxoplasma retinochoroiditis	-	1st dose	8	blurred vision	20/40	20/20	sulfadiazine and pyrimethamine tablets, oral prednisone	complete resolution
12	LE	Toxoplasma retinochoroiditis	Toxoplasma retinochoroiditis	2nd dose	7	blurred vision	20/200	20/20	sulfadiazine and pyrimethamine tablets, oral prednisone	complete resolution
13	RE	Toxoplasma retinochoroiditis	Toxoplasma retinochoroiditis	1st dose	7	blurred vision	20/50	20/20	sulfadiazine and pyrimethamine tablets, oral prednisone	complete resolution
14	BE	VKH disease	VKH disease	2nd dose	12	blurred vision	RE: 20/22 LE: 20/25	RE: 20/20 LE: 20/20	MMF 2g, oral prednisone	complete resolution
15	BE	VKH disease	VKH disease	2nd dose	5	blurred vision	RE: 20/25 LE: 20/28	RE: 20/20 LE: 20/22	MMF 2g, oral prednisone	complete resolution
16	BE	pars planitis	-	1st dose	7	blurred vision	RE: 20/25 LE: 20/20	RE: 20/20 LE: 20/20	oral prednisone	complete resolution
17	BE	pars planitis	-	2nd dose	14	blurred vision	RE: 20/20 LE: 20/20	RE: 20/20 LE: 20/20	oral prednisone	complete resolution
18	BE	retinal vasculitis	-	2nd dose	5	blurred vision	RE: 20/66 LE: 20/20	RE: 20/20 LE: 20/20	oral prednisone	complete resolution
19	RE	retinal vasculitis	-	1st dose	10	blurred vision	20/28	20/20	oral prednisone	complete resolution
20	BE	panuveitis in new-onset BD	-	2nd dose	30	redness, blurred vision	RE: 20/28 LE: 20/32	RE: 20/25 LE: 20/22	oral prednisone, AZA	complete resolution
21	LE	MEWDS	-	2nd dose	28	decreased VA, visual field defect	20/25	20/20	-	complete resolution
22	RE	MEWDS	-	1st dose	4	blurred vision, visual field defect	20/66	20/20	-	complete resolution
23	RE	MEWDS	-	1st dose	7	decreased VA	20/400	20/20	-	complete resolution
24	BE	AMN	-	1st dose	2	visual field defect	RE: 20/20 LE: 20/20	RE: 20/20 LE: 20/20	-	significant improvement
25	RE	CRVO	-	2nd dose	30	decreased VA	20/400	20/100	intravitreal anti-VEGF	mild improvement
26	LE	BRVO	-	1st dose	2	decreased VA	20/100	20/40	intravitreal anti-VEGF	partial improvement
27	LE	BRVO	-	2nd dose	2	decreased VA	20/32	20/25	intravitreal anti-VEGF	partial improvement
28	LE	BRVO	-	2nd dose	3	decreased VA	20/22	20/20	intravitreal anti-VEGF	significant improvement
29	LE	BRVO	-	2nd dose	23	blurred vision	20/20	20/20	intravitreal anti-VEGF	significant improvement
30	RE	NAION	-	1st dose	2	decreased VA, visual field defect	20/40	20/200	oral prednisone	no improvement
31	RE	uveitic CNV	Toxoplasma retinochoroiditis	1st dose	8	decreased VA	20/200	20/40	intravitreal anti-VEGF	significant improvement
32	RE	uveitic CNV	Serpiginous Choroiditis	2nd dose	10	decreased VA	20/50	20/32	intravitreal anti-VEGF	partial improvement
33	RE	myopic CNV	-	2nd dose	1	blurred vision	20/32	20/25	intravitreal anti-VEGF	partial improvement
34	BE	CSR	-	2nd dose	13	blurred vision	RE: 20/22 LE: 20/50	RE: 20/20 LE: 20/22	-	complete resolution

BE: both eyes; LE: left eye; RE: right eye, NGAU: non-granulomatous anterior uveitis; HLA-B27: human leukocyte antigen B27; CMV: Citomegalovirus; AU: anterior uveitis; VKH: Vogt-Koyanagi-Harada; BD: Behçet’s disease; MEWDS: multiple evanescent white dot syndrome; AMN: acute macular neuroretinopathy; CNV: choroidal neovascularization; CRVO: central retinal vein occlusion; BRVO: branch retinal vein occlusion; NAION: non-arteritic ischemic optic neuropathy; CSR: central serous retinopathy; VA: visual acuity; BCVA: best corrected visual acuity; MMF: mycophenolate mofetil; AZA: Azathioprine; -: none; VEGF: vascular endothelial growth factor.

## Data Availability

Not applicable.

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
