# Peer review of "Uveitis and Other Ocular Complications Following COVID-19 Vaccination"

_jcm, 2021, doi:10.3390/jcm10245960_

Round 1
Reviewer 1 Report
This is a very interesting case-series about ocular inflammatory diseases and their relapses that is highly interesting for the medical community. There are only minor comments from my side:
The abstract includes a long general introduction/background information that can be shortened but lacks an overview of the reported case series (how many patients, mean age, females/males, type of ocular diseases, etc).
paragraph 3.3: please provide commercial vaccination name or company (although already mentioned in the introduction), as most readers will not be able to identify the vaccination just by their ingredient code.
Table 1: As not all patients had an uveitis attack, the column „Time interval between the last uveitis attack to current uveitis (years) has to bere-written. Furthermore, it is unlikely that the recurrences occurred exactly 1 or 2 years after the last episode. Please provide the time interval either in years with one decimal place, or otherwise in months.
The conclusion should summarize the findings and the potential underlying pathomechanisms. I would move the limitations of the study from the conclusion to the discussion section.
The authors may include some thoughts about potential prophylactic treatment in patients with known uveitis to prevent recurrence due to vaccination.
Author Response
The abstract has been shortened and an overview of the reported case series (how many patients, mean age, females/males, type of ocular diseases, etc) has been added.
Paragraph 3.3 now includes the commercial vaccination names or company.
In Table 1, the time interval between the last uveitis attack to current uveitis (years) has been re-expressed in months.
In the Conclusion, the findings and the potential underlying pathomechanisms have been summarized.
The limitations of the study have been moved from the Conclusion to the Discussion section.
Remarks regarding potential prophylactic treatment in patients with known uveitis to prevent recurrence due to vaccination have been included.
Reviewer 2 Report
The authors showed the cases of possible ocular complications after COVID-19 vaccination and compared with previous reports. Since most people are vaccinated these days, they happen to develop ocular diseases after vaccination that are not related to the vaccine. Thus, it is difficult to define ocular diseases are associated with the vaccination. While accumulation of these complication is important in COVID-19 pandemic, some complications suggested by the authors do not seem associated with the vaccination.
For example, Patient 11 who had no history of ocular toxoplasmosis developed toxoplasmosis after the vaccination. I do not think this infection was caused by the vaccination. It seems that Patient 11 happened to be infected with Toxoplasma at the time he was vaccinated. For Patient 10, it is also difficult to show that CMV infection was caused by the vaccination. It is difficult to deny that chronic CMV anterior uveitis was happened to be diagnosed after the vaccination.
The cases of BRVO or CRVO seems difficult to prove the complication of the vaccination. Is there any specific feature (clinically or laboratory data) comparing to those not associated with the vaccination?
Other comments are followed;
- Table 1 and 2 are related, but it is difficult to correlate each patient data. The author need to edit tables, at least Table 2 needs data of past uveitis history.
- Patient 34 developed CSR, which was resolved by oral steroid. Generally, systemic steroid is contraindication for CSR. Why was this patient treated with oral steroid? The authors need to clarify this point.
Author Response
We agree with the reviewer that ocular toxoplasmosis cannot be caused by the vaccination. In patient 11, the vaccination might have caused toxoplasmosis reactivation. Similarly, patient 10 had no history of CMV uveitis, but the COVID-19 vaccination may have triggered its reactivation. What this and other papers suggest is that COVID-19 vaccination may trigger infectious reactivation. This has been better outlined in the paper.
Cases of BRVO or CRVO are difficult to prove as complications of the vaccination. However, a close association in young patients, some even without other risk factors, cannot be ruled out. Unfortunately, there are no specific features (clinically or laboratory data) compared to those not associated with the vaccination. So far, only changes affecting retinal vessels during the acute phase of COVID‐19 and after patients’ recovery have been studied. The study assessed whether COVID-19 can induce important changes to the retinal vasculature during the acute phase of the disease, including microvascular infarcts and major artery and vein dilation. Most of these alterations disappear six months after the disease resolution, suggesting a possible correlation with the generalized inflammatory and pro-coagulant status typical of acute COVID-19. Further studies are needed to investigate the actual correlation between retinal alterations and thrombotic risk in patients after COVID-19 vaccination.
Girbardt C, Busch C, Al-Sheikh M, Gunzinger JM, Invernizzi A, Xhepa A, Unterlauft JD, Rehak M. Retinal Vascular Events after mRNA and Adenoviral-Vectored COVID-19 Vaccines-A Case Series. Vaccines (Basel). 2021 Nov 17;9(11):1349. doi: 10.3390/vaccines9111349. PMID: 34835280; PMCID: PMC8625395
Bialasiewicz AA, Farah-Diab MS, Mebarki HT. Central retinal vein occlusion occurring immediately after 2nd dose of mRNA SARS-CoV-2 vaccine. Int Ophthalmol. 2021 Dec;41(12):3889-3892. doi: 10.1007/s10792-021-01971-2. Epub 2021 Aug 23. PMID: 34426861; PMCID: PMC8382109.
Endo B, Bahamon S, Martínez-Pulgarín DF. Central retinal vein occlusion after mRNA SARS-CoV-2 vaccination: A case report. Indian J Ophthalmol. 2021 Oct;69(10):2865-2866. doi: 10.4103/ijo.IJO_1477_21. PMID: 34571653.
- We completely agree that the two tables are related and that it is difficult to correlate each patient’s data. We previously tried to merge the two tables, but, unfortunately, there are too many data to fit in just one. As suggested by the reviewer, the data of past uveitis history have been added in Table 2.
- Patient 34 was referred to our center for suspected VKH disease, and treatment with oral prednisone was given at presentation. Afterward, prednisone was suspended. In accordance with the reviewer, we have deleted “prednisone” in order to avoid misinterpretation.
Round 2
Reviewer 2 Report
The edited article was addressed the reviewer's comments.